# A behavioral economics analysis of the participation in early childhood development social programs promoted by civil societies in Mexico

Edson Serván-Mori[1‡], Carlos Pineda-Antúnez[1‡], María L. Bravo-Ruiz[1], Mariana Molina[2], Martín I. Ramírez-Baca[1], Angélica García-Martínez[3], Amado D. Quezada-Sánchez[4], Emanuel Orozco-Núñez[1]*

1 Center for Health Systems Research, National Institute of Public Health, Cuernavaca, Mexico, 2 Faculty of Social Sciences, McMaster University, Hamilton, Canada, 3 Lucy Family Institute for Data and Society, University of Notre Dame, Notre Dame, Indiana, United States of America, 4 Center for Evaluation and Surveys Research, National Institute of Public Health, Cuernavaca, Mexico

‡ ESM and CPA contributed equally to this work and share first authorship.
* emanuel.orozco@insp.mx

**Data Availability Statement:** Interviews were conducted, transcribed and analyzed in Spanish language. Field interviews and focus groups were

## Abstract

Based on a behavioral economics (BE) approach, we analyzed the decision to participate in an early childhood development (ECD) program implemented in Mexico by a non-governmental organization. We conducted a literature review and a qualitative study of four localities participating in the ECD program. Situated in the state of Oaxaca, these communities are characterized by high and very high levels of social marginalization. From May 20 to 30, 2019, we collected primary data through semi-structured interviews (n = 30) and focus groups (n = 7) with a total of 61 informants (51 women and 10 men). We then performed an inductive systematic analysis of the data to identify documented cognitive bias associated with the decisions of individuals to participate and remain in or abandon social programs. The interviewees were living in conditions of poverty, facing difficulties in meeting even their most basic needs including food. Program participants attached far greater weight to incentives such as the basic food basket than to the other benefits offered by the program. The four localities visited maintained traditional views of domestic roles and practices, particularly regarding child-rearing, where women were in charge of childcare, home care and food preparation. Problems linked to child malnutrition were a decisive factor in the decision of residents to participate and remain in the program. Testimonials gathered during the study demonstrated that the longer the mothers remained in the program, the more they understood and adopted the concepts promoted by the interventions. In contexts marked by economic vulnerability, it is essential that ECD programs create the necessary conditions for maximizing the benefits they offer. Our analysis suggests that cognitive load and present bias were the biases that most severely affected the decision-making capacity of beneficiaries. Therefore, considering loss aversion and improving the management of incentives can help policymakers design actions that "nudge" people into making the kinds of decisions that contribute to their well-being.

translated verbatim. After that, a team of three researchers using the Atlas-Ti software and 32 codes, created thematic data sets integrated in a unique qualitative electronic database. Our informed consent approved by our Ethics Committee establish that identity of informants will be kept under secrecy. For data valilability, please contact to Dr. Ileana Heredia (ileana.heredia@insp.mx), Ethics Committee Representative of the Centre for Health Systems Research of the National Institute of Public Health of Mexico. If some concern persists about our ethical considerations, please contact Dr. Angelica Angeles Llerenas (aangelica@insp.mx), President of the Ethics Committee of the National Institute of Public Health of Mexico.

**Funding:** This work was possible with the support of the Non-governmental Organization Un Kilo de Ayuda A.C., Mexico (UKA). The funder had no role in study design, data collection and analysis, decision to publish, or preparation of the manuscript.

**Competing interests:** The authors have declared that no competing interests exist.

**Abbreviations:** BE, Behavioral Economics; BIAS, Behavioral Interventions for Self-Sufficiency; CDP, Community Development program; CEDIT, Early Childhood Development Centres; ECD, Early Child Development; MIDIT, Comprehensive Model of Early Childhood Development; NGO, Non-Governmental Organization; NPECDP-UKA, *UKA's* Neurological and Psycho-affective Early Childhood Development Program; UKA, *Un Kilo de Ayuda, A. C.*.

## Introduction

Social programs in low- and middle-income countries (LMICs) are normally designed assuming that individuals actively make decisions and complete all the steps required to benefit from their contributions. They take it for granted that people decide which programs are suitable for them, fill out forms, attend meetings and carry out all the specified procedures. Program designers often suppose that individuals carefully consider all the options, analyze details and make decisions that optimize their wellbeing [1]. However, in contrast to standard economic theory, behavioral science experiments have shown that behavior deviates from standard theory at each stage of the decision-making process, that is, people have 1) non-standard preferences, 2) erroneous beliefs and 3) systematic biases [2]. As a result, participation in social programs often proves lower than expected, even when the benefits outweigh the costs of participating [1].

It is generally believed that participation or abandonment is the result of an implicit decision whose determinants and implications are in most cases ignored. The decision to participate in a social program is part of a mental process molded not only by the way the program is designed and implemented and the type of follow up provided, but also by the feedback participants receive on their activities [3]. It is also shaped by the social environment of participating communities including the judgments, values, motivations and principles of the population involved [4, 5]. These element bears implications for the effectiveness of social programs [6, 7]. It has been suggested, for instance, that having incomplete information or underestimating about the future benefits of a program can inhibit the participation of potential beneficiaries [8].

Behavioral economics (BE) serves to analyze decisions in depth and thus to anticipate ways of improving programs. This field of economics highlights the relevance of incorporating decision and behavioral models into the design and implementation of public policy [9, 10], bearing in mind that all individuals are subject to cognitive biases when making decisions. BE attaches particular importance to 1) the role of emotions in the formation of judgments and the decision-making process; 2) the competing claims of altruism vs. self-interest; 3) self-control and its limits; and 4) the human biases and limits experienced when assessing future benefits, as well as the difficulty of making inter-temporal decisions [11–14]. BE offers useful tools for analyzing decision making and its implications for public policy, taking into account key elements such as automatic thinking, social thinking and thinking based on mental models [15, 16]. It maintains that the timely identification of cognitive biases can contribute to the success of interventions.

Although research and overall interest in behavioral economics and psychology have grown by leaps and bounds in recent decades, a wide gap in the literature remains regarding the factors underlying the decision of community members on whether or not to participate and continue in social programs aimed at improving early childhood development (ECD) [17, 18]. Little evidence exists on the behavioural correlates of such decision making among individuals in contexts marked by acute social deprivation in LMICs such as Mexico. This is particularly relevant given that children–male and female–are more socially vulnerable than other population groups in these contexts, where the enforceability of their rights depends on the decisions and preferences of others. We conducted a brief literature review on BE in relation to the design of social programs and examined the ways in which its conceptual components could be used to analyze the decisions of individuals regarding whether or not to join and remain in such programs. This paper specifically analyzes the decisions of residents of poor communities in Oaxaca, in Mexico, who have participated in the ECD programs implemented by the Mexican non-governmental organization (NGO) *Un Kilo de Ayuda* (*UKA*). Finally, this work ponders the implications of our findings for the design, implementation, monitoring, and performance of ECD programs.

## Brief literature review

BE proposes decision-making models based on the analysis of a mental process that combines psychology with economics [19]. The BE models are anchored in the theoretical contributions of Kahneman [20] and Thaler & Sunstein [21] regarding the mechanisms and factors that influence decisions and the process that precedes them. These authors hold that traditional microeconomic assumptions are limited in their explanation of the individual decision-making process [20, 21]. Contrary to the traditional assumptions, they propose that decision making is influenced by context, temporality, consequences and prior knowledge of the implications of the decisions [22].

Several empirical studies have explained the ways in which BE has influenced the design of policies aimed at boosting enrollment in retirement savings programs, increasing organ donations and expanding influenza vaccination. These authors have analyzed the factors underlying the cognitive process of decision making, among them procrastination, the perceived complexity of tasks and inadequate planning [23]. After reviewing various government initiatives intended to motivate self-care in cases of chronic disease, the adoption of healthy lifestyles, and adherence to medication and medical follow up, Matjasko et al. revealed that incorporating small "nudges" into the design of interventions produced positive outcomes [24].

Several studies have also suggested that, apart from the cognitive process of decision making, exogenous elements influence the determination of individuals to participate and remain in or abandon social programs. Salient among these elements are the value judgments of external agents such as the individuals who administer the programs. This applies to social programs which, similarly to the *UKA* interventions analyzed in our study, seek to improve the well-being of children by educating their parents. These authors indicate that the *framing bias* in the design and implementation of social programs outweighs the other biases identified by BE, as shown before [25–28].

The importance of incorporating the BE perspective into the design and implementation of social programs has been widely recognized. For instance, the project *Behavioral Interventions to Advance Self-Sufficiency (BIAS)* in the United States has demonstrated the benefits of implementing child-centered policies by incorporating "nudges" regarding the well-being of children. The authors of this project described the cost-effectiveness of considering biases such as *social influence*, *personalization* and *loss aversion* when designing a child-care intervention. Their results indicated improvements of two-to-four percentage points in the indicators analyzed, namely keeping medical appointments, establishing commitments with social assistance programs and requesting educational credits during appointments [29]. Other studies have shown that using BE elements such as *present bias*, *temporal preferences*, *loss aversion* and *context* can accelerate improvements in maternal and child health [30–32]. Reviewing programs and interventions from a BE perspective allowed for identifying the exact points in time when parents require assistance in making favorable decisions. In turn, this made it possible to identify windows of opportunity not only to improve the timing, contents, and forms of communication with parents, establishing an alert mechanism for tracking key phases, but also to simplify procedures. However, most importantly, BE enabled us to identify which improvements would contribute to decision making, knowing that people make decisions based on limited information and not necessarily at the right time or in ideal contexts.

Experimental evidence has shown that parents in interventions that apply BE principles are more likely to participate and remain in the programs of interest, as well as to comply with follow-up activities for their children [33]. Interventions that have endeavored to maintain the interest of parents by mitigating behavioral barriers that prevent greater commitment to their children have achieved better results than if they had viewed the parents merely as vehicles of

information. York and Loeb (2018) evaluated a reading program, finding that children whose parents believed in the benefits of the program obtained better results than those whose parents did not [34]. Similarly, Mayer et al. (2018) showed that the provision of immediate rewards to parents contributed to achieving continuity in intervention activities [35]. Recent evidence suggests that for parents to remain in social programs and achieve long-term changes in activities geared to the needs of their children, it is essential to take into account their barriers, for instance, as regards time, language and expectations [33]. In the case of minority populations, it has been observed that "nudges" lose relevance as motivators of permanence where participating parents do not identify with the interventions [36].

## Conceptual elements

**Cognitive bias and participation in social programs.** Neoclassical microeconomic theory assumes that individuals are fully informed, make optimal and rational decisions, have well-defined preferences and calculate choices rapidly before making a decision [37]. However, these assumptions are not always correct. Since the advent of BE, a growing body of evidence has indicated that cognitive biases are often associated with suboptimal decision making. In economics, irrational decisions are those that do not maximize utility, thereby causing a loss of economic welfare. BE is particularly interested in exploring why people make irrational decisions and why their behavior deviates from the economic models explaining the association between the dominant choice among people in society and cognitive bias [21].

Cognitive biases relate to the amount of mental resources employed at a given moment [38] which affect basic skills such as attention, cognitive capacity and execution control [39]. They also concern the manner in which people reason and resolve problems. Many decisions are dictated by the nature and cognitive biases of the individuals who make them. This is not to say that people are irrational, but rather, that they make systematic errors that translate into suboptimal decisions [40].

Moreover, it has been shown that parents living in contexts of deprivation are permanently burdened by a high cognitive load that limits their capacities and affects their ability to control their decisions [41]. This constrains their awareness as to the consequences of their decisions for the future development of their children [35].

ECD programs are permanently faced with the challenge of achieving the enrollment and ensuring the adherence of eligible individuals and families. BE offers analytical tools for elucidating low participation levels [42, 43] which could have their origins in a range of issues such as difficulties in understanding the costs and benefits of choices or in sorting out a large number of options; a preference for immediate over future benefits; a misconception of the magnitude of the risk involved; or problems associated with the way people frame their problems in their minds. Table 1 presents a set of cognitive biases related to participation in social programs.

**Elements related to the way social programs are offered.** The search for strategies that help mitigate cognitive biases and generate optimal decision making is a fundamental part of designing and implementing social programs. ECD programs involving the voluntary participation of beneficiaries face a particularly arduous challenge, since they must not only ensure that the information provided to parents and caregivers will positively impact the development of children, but also maintain the interest of participants in the program. Since the advent of BE, it has become increasingly clear that the role of organizations transcends the provision of services or goods. To obtain the desired results, they must pay attention to the way they prepare the information, considering the contextual determinants of the decision-making process that potential participants will face.

**Table 1. Cognitive biases that influence participation in social programs [3, 21, 22, 45, 59].**

---

***Present bias.*** It is natural for people to value the present moment more than the future. This preference, known as temporal discounting, is illustrated by the fact that people obtain pleasure from eating fatty and sweet foods knowing that doing this persistently will cause obesity, diabetes, and other diseases. The basic problem here is self-control, especially when current temptation ("that chocolate") is considered especially important while the future implications for health are remote and gradual. A favorite approach of behavioral economists to mitigate such temptation is to eliminate it; they have recognized that farsightedness is outweighed by the tangible presence of the seductive element.

---

***Limited attention.*** Where possible, individuals prefer to process information rapidly and through intuitive thinking, reserving deliberative thinking for special situations. The need to evaluate many options can also discourage people in that they feel overwhelmed with options.

---

***Decision fatigue.*** When people make several successive decisions, they become mentally exhausted and begin making inconsistent and poorly calculated decisions.

---

***Prospect theory.*** Contrary to the assumptions of neoclassical theory, behavioral economics affirms that people evaluate risky choices according to the gains and losses–rather than the utility–associated with their outcomes.

---

***Risk aversion.*** Experimental studies have demonstrated that people are twice as sensitive to losses as they are to gains.

---

***Heuristics.*** Rather than undertaking a decision-making process that maximizes utility based on the consideration of all available information (one of the assumptions of neoclassical theory), people follow practical rules or mental shortcuts for making decisions.

---

***Commitments.*** People tend to delay decision making when their interest in obtaining an objective is not short-term. Many are aware of having weak will power (e.g., they tend to spend too much, eat excessively or continue smoking). It has been demonstrated that commitments support the completion of short-term objectives.

---

***The influence of messengers.*** People are influenced by those who provide information. There is ample evidence to the effect that the perception of authority tends to generate compatible behavior, even if such behavior is stressful or detrimental.

---

***Incentives.*** A large body of research has shown that the use of cash or in-kind incentives can motivate people to change their behavior, for instance, to eat healthier food, do more exercise, consume fewer alcoholic drinks or stop smoking.

---

***Social norms.*** It is the internalized social and cultural norms that dictate for individuals how they should behave. These norms function as rules in societies or communities to which the members of a social group attempt to adapt. They can be "descriptive norms" which depict how individuals tend to behave (e.g., "the majority of men are providers"), or "prescriptive norms" which establish what is acceptable or desirable behavior (e.g., "people should arrive at work on time").

---

***Prominence.*** People are more likely to fix their attention on things they can understand or easily codify. Presenting information as simply as possible helps maintain the attention of listeners more than presenting it in an abstract manner.

---

***Priming.*** People behave differently when they have been "prepared" prior to undertaking a task by exposing them to certain signals or stimuli. The impact of priming is unconscious.

---

Table 2 presents several strategies commonly implemented to increase participation and permanence in social programs. They include activities such as exerting influence on the key sectors of the beneficiary population in order that, in turn, they may generate a change in the social behavior norms of the community [44]; enhancing the coverage and relevance of program benefits such that caregivers may feel motivated to participate; stepping up efforts to reach families in remote areas [43]; and incentivizing participation by reaching commitments with caregivers as a means of combating procrastination and absenteeism [45]. Among the most documented strategies are the use of reminders (e.g., via text messages) and simplified program processes, both of which seek, on one hand, to reduce the cognitive load involved in completing actions, and on the other, to avoid fatigue among participants while they decide to either continue or stop attending program activities [7, 46]. In addition, it has been documented that decisions are strongly affected by the way the information is presented. This highlights the value of adopting strategies for framing information in a manner that attracts the largest possible number of potential beneficiaries to the program [39]. Finally, the use of financial incentives is perhaps the strategy most frequently used by social programs; its effectiveness

**Table 2. Strategies identified in the literature for promoting participation in social programs.**

| Strategy | Definition | Recommendation | Source |
|---|---|---|---|
| Influencing the population | Exerting influence on a key sector of society leads to increased participation. | Identify the "push" and attraction factors that contribute to positive behavioral changes and transfer the positive practices in anomalies to social norms. | [44] |
| Establishing commitments with caregivers | Committing beneficiaries helps avoid procrastination in the performance of activities. | Establish explicit commitments with deadlines for key activities. Where possible, personalize the commitments. | [45] |
| Designing simple program options | Simple choices and procedures help prevent demotivation caused by decision fatigue. | Offer participants clear and simple information; | [46] |
|  |  | Ensure that program enrollment and adherence procedures are easy to follow; and |  |
|  |  | Do not overload beneficiaries with excessive processes or registration and compliance procedures. |  |
| Sending reminders | Reminders help reduce the cognitive load experienced in completing an action. | Send reminders to participants with the dates and times of future activities; | [7] |
|  |  | Reminders are also useful for indicating whether there are any tasks pending and for motivating participants with information on their progress. |  |
| Framing information adequately | Choices are also influenced by the way they are framed. People prefer information presented in a positive rather than a negative manner. | Present the benefits that can be obtained through the program from a positive angle; for instance, it is preferable to affirm "Eating adequately prevents malnutrition," than to state "Not eating adequately causes malnutrition." | [39] |
| Providing financial incentives | Contributing to the income security of beneficiaries reduces their cognitive load. | Providing financial incentives can have an indirect effect on the development of children inasmuch as it reduces psychological stressors that cause parents to miss sessions or refrain from committing themselves positively with their children. | [41] |

resides in generating a sense of security among potential beneficiaries regarding their income, or at least in countering the idea that their participation might undermine their income. This strategy reduces the cognitive load of caregivers and thus contributes to the establishment of commitments with the children [41].

**Structural factors.** The decision to participate in a social program does not take place in a vacuum, nor does it respond exclusively to individual interests. It evolves in a space delineated by social features such as the stage of development, available infrastructure, local power structures, traditional customs and habits, and level of geographic isolation of communities, together with other contextual characteristics that are largely immune to modification by individuals.

Structural determinants represent another window of opportunity for generating behavioral changes [47]. It is important to identify solutions that mitigate social vulnerability and boost the resilience of marginalized societies by means of policies on urban planning, technological advancement, and business development, among others. Even though some of these factors may not directly affect the decision of individuals to participate in social programs, they *do* influence their cognitive biases. This is especially the case in environments marked by low levels of development, hard-to-access areas and inadequate health and educational resources [16]. Poverty prevents people from easily internalizing the benefits of investing time or resources in projects. Implementing strategies such as those proposed by BE acquires special relevance in these circumstances: it has been demonstrated that once a critical mass of community members adopts a new (e.g., healthier) behavior, barriers stemming from deep-seated customs and habits dissolve. This facilitates sustainability [44].

**Decision making in contexts of poverty.** Poverty increases the cognitive load of individuals and limits their resolve to act according to their desires [41, 48]. In the field of psychology, there is consensus concerning the presence of two systems of thought: the first, reflexive and

deliberative, calls for greater effort and is therefore more difficult to develop; the second is intuitive and requires no cognitive effort. Thaler and Sunstein designated the first as a reflexive and the second as an automatic thought system [21]. In contexts of limited material and cognitive resources, information is inadequately assimilated, and more decisions are made under the automatic than under the reflexive thought system used by individuals who are not suffering from deprivation [41]. Decisions made under the automatic thought system regarding benefits that unfold over the long term tend to be suboptimal because this system is more vulnerable to the influence of decision-making biases; these biases are directly or indirectly associated with the participation of parents in social programs [16].

## Materials and methods

Our study was nested within a broader project aimed at assessing the impact of *UKA*'s Neurological and Psycho-affective Early Childhood Development Program (NPECDP-*UKA*) in several marginalized municipalities of Oaxaca, Mexico.

### The intervention

*UKA* was created over three decades ago with the purpose of contributing to the eradication of childhood malnutrition. The organization operates through eight Early Childhood Development Centers (*CEDITs* for its acronym in Spanish) which up to 2018 had served over 30 000 children under five living in areas with high and very high levels of social marginalization.

It operates in five Mexican states: Chiapas, the State of Mexico, Oaxaca, Sinaloa and Yucatan. Its actions, based on the Comprehensive Model for Early Childhood Development (*MIDIT* for its acronym in Spanish) (Fig 1), are centered primarily on helping children reach their full development through three subprograms: 1) the Physical Development Program (PDP) provides monitoring and nutritional status assessment based on weight, length and height measurements that serve to estimate undernutrition, overweight and obesity; it also implements capillary blood sampling for anemia diagnosis; 2) the NPECDP-*UKA* offers neurological assessment based on ECD testing [49, 50] and provides caregivers with counseling and workshops on early childhood stimulation; and 3) the Community Development Program (CDP) contributes to food security by offering food packages at affordable prices and

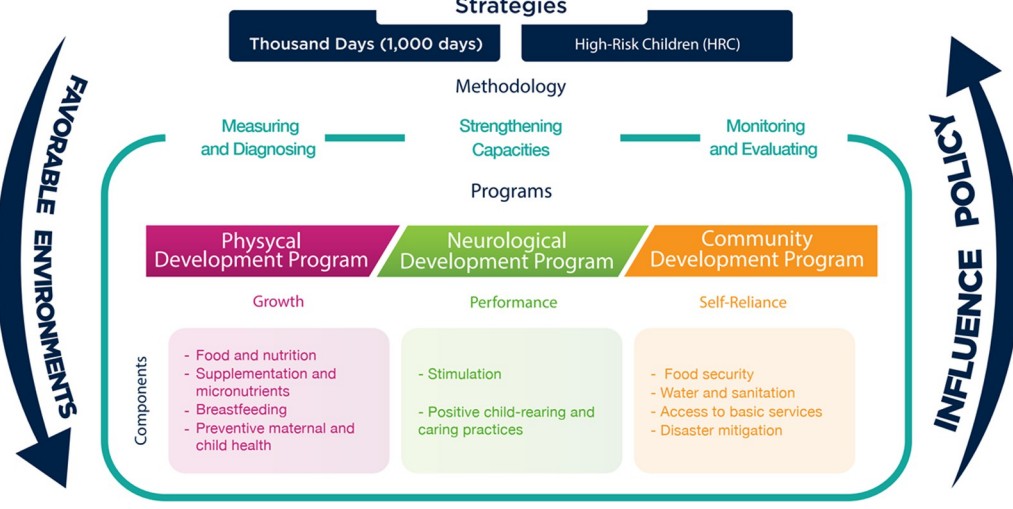

**Fig 1. Comprehensive model for early childhood development (*MIDIT*).**

promoting family orchards. To enhance the knowledge and skills of caregivers, these programs hold workshops on five central themes in public community spaces: child-rearing, breastfeeding, food supplements, undernutrition, and maternal and child health.

*MIDIT* participation consists of six phases (Fig 2): 1) <u>Eligibility</u>: during this phase, *UKA* staff and local authorities jointly select the communities where the program will be implemented throughout the year. Kickoff of the *UKA* intervention is announced in the communities, and residents are informed that the benefits are intended for families with children under five. 2) <u>Knowledge</u>: Community members obtain information about the intervention through either the program outreach strategy or word-of-mouth promotion from individuals who have already been informed. 3) <u>Application</u>: those who are interested in participating complete forms and submit the required documentation including a community consent sheet. 4) <u>Acceptance</u>: *UKA* staff inform applicants of their acceptance into the program. The preceding four phases converge during the process of outreach and awareness meetings. The sessions include discussions on the *MIDIT*, its programs, its benefits, and the requirements for obtaining them. 5) <u>Registration</u>: The entry of each applicant into the program is formally logged into a register of beneficiaries. This phase takes place during the third or fourth visit from *UKA* staff. At this point, weight, length and height measurements are taken and capillary hemoglobin samples are collected in accordance with the operative plan. 6) <u>Follow up</u>: The mothers/caregivers who decide to participate in the program and confirm their decision by registering can then move on to the actual participation phases. All participating families receive the non-conditional program benefits on a monthly basis and are free to abandon follow up at any time.

## Study sites and participants

With a population of almost four million (of whom 31% are indigenous) located in the southwest region of Mexico, Oaxaca is one of the most socially disadvantaged states in the country. Life expectancy at birth falls below the national average [51], and in 2018, 66% of the population lived in poverty, with only 16% enjoying access to social security while 27% exhibited educational gaps [52].

For our study, we selected four localities with high and very high levels of social marginalization served by the *UKA* organization: San Antonino el Alto, Magdalena Yodocono de Porfirio Diaz,

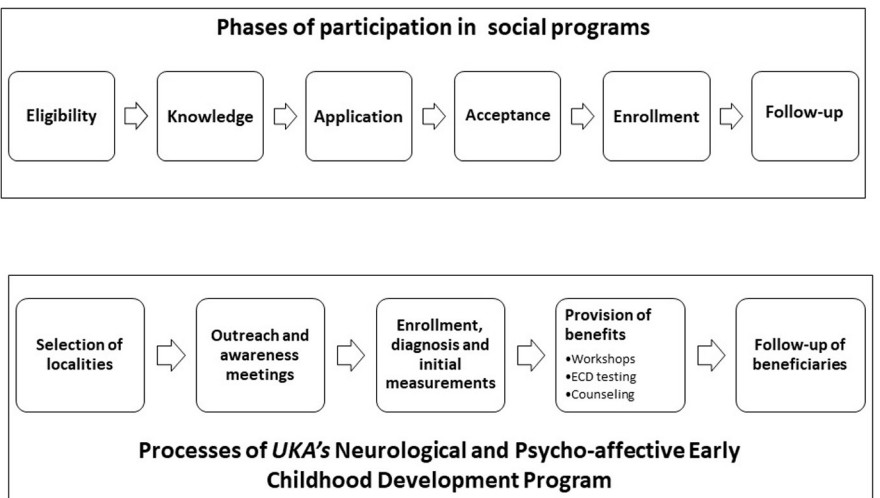

**Fig 2. Phases of participation and permanence in the *MIDIT*. Source:** Elaborated by the author based on the proposals of Heckman (2004) [9].

Rancho San Felipe (Santiago Matatlán) and San Simon Almolongas. Located 30–70 kilometers from the state capital, the inhabitants of these towns are engaged primarily in the production and distribution of agricultural products, livestock raising, retail trade, construction work and domestic work. From May 20 to 30, 2019, we conducted 27 semi-structured interviews and seven focus groups with 61 participants (51 women and 10 men). The field sample is presented in Table 3.

To achieve our field objectives, we formed a purposive sample of program participants with the collaboration of *UKA* personnel who identified and recruited informants during their monthly visits. Once having completed our sample, we agreed on convenient dates for conducting interviews. Three groups of respondents were identified. The first included the "committed" parents of children in the program, those who believed that it was important to participate in the *MIDIT*. They were comprised of the program "commissioners," or mothers who also worked as translators/interpreters and were engaged in *UKA* operations, as well as their partners (n = 12). The second group included "demotivated" parents, those who joined but later abandoned the program (n = 7 in total). The third group of interviewees included operational staff: the *CEDIT* manager, NPECDP-*UKA* coordinator and *MIDIT* facilitators (n = 8 in total). We interviewed this group of individuals to understand the context in which beneficiaries participated and to obtain information on the operational processes affecting participation. Finally, we conducted seven group interviews: six with committed participants and one with *UKA* personnel.

## Interview guides

We designed four interview guides based on the above-mentioned literature review (available in https://doi.org/10.6084/m9.figshare.14601726.v1). Personal, contextualized data from participants were explored thematically. Knowledge concerning *UKA* operational processes was also explored. This included eligibility criteria, motivations and external influences, knowledge of expected program benefits, costs and barriers to participation, knowledge concerning childhood growth and development, leadership and decision making within the home, gender roles in child-rearing and a report on the level of satisfaction with participation in the *UKA* programs. The guides were piloted and implemented by experienced individuals. Interviews with *UKA* staff were conducted in their offices.

## Validity and rigor

The credibility and consistency of our data were ensured through interpretive triangulation, which involved the participation of several researchers in the analysis and interpretation of data [53]. Confirmability and validity of our data were established by comparing the themes of our findings and concepts analyzed against the literature. We also held critical reflection sessions among team members in order to ensure agreement on the research problem, theory and methodology [54].

## Analysis of interviews

We conducted an inductive analysis following agreed upon, systematic steps [55]. Interviews were transcribed, substituting codes for participant names and erasing the audio

**Table 3. Participants in field interviews.** Oaxaca, México, 2019.

| Type of interview | Individual Interviews | | | Group Interviews | |
|---|---|---|---|---|---|
| Type of informant | Commited | Non Commited | *UKA* personnel | Commited | *UKA* Personnel |
| Number of participants | 12 | 7 | 8 | 53 | 8 |

Source: Proper elaboration.

files once transcriptions had been safeguarded. Texts were organized into an electronic file for analysis using the Atlas-Ti software, Version 7 [56]. This process emulated the practice of segmenting and organizing the diverse contents of the texts by theme for subsequent analysis. The field team defined these codes based on the data collected, making sure that the information was as specific as possible. We used the free coding tool from our software package to facilitate the classification of the information into the following codes: *commissioners, doubts and comments, participant profiles, attitudes towards attending/abandoning the program, eligibility, motivation of commissioners, attention skills, measurement, perceived benefits, UKA reputation, program implementation provision of food package, decision making, involvement of partner, outreach, talks and workshops, concerns, expected benefits, educational level, growth and development, satisfaction, abandoning the program, child-rearing skills, occupation, general information, household composition, continuing with the program, intervention, gender roles, improvements, inconveniences, enrollment and motivation.*

We coded the interviews by respondent group after reaching a consensus among research team members and reviewed the coding in pairs for quality-control purposes. We then created a file of all codes and corresponding text segments. We used the classification of cognitive biases described above to provide theoretical support for the data presented in the results section.

## Ethical considerations

Following the ethical principles outlined in the Helsinki Declaration for Medical Research Involving Human Subjects, we obtained informed consent from each participant, having first explained the consent section of the questionnaire [57]. Those who agreed to participate were asked for their authorization to tape record their interviews. This study was approved by the Research, Biosafety and Ethics Committees of the National Institute of Public Health in Mexico (ID: 1649–7151). The research protocol of the program evaluation was also approved by the same committees (CI-896-2018/1538) and its experimental component registered in ClinicalTrials.gov (ID: NCT04210362).

## Results

Our analyses allowed us to identify eight biases or cognitive barriers associated with the decision of beneficiaries to participate, remain in, or abandon the program: 1) *cognitive load;* 2) *present bias and incentives;* 3) *social norms;* 4) *availability of information;* 5) *simplicity of the process and influence of intervention facilitators;* 6) *loss aversion;* 7) *commitments* and 8) *status quo bias.* Table 4 presents these biases by frequency.

## Cognitive load

Regardless of the type of program participant, the impoverished circumstances of interviewees–with the consequent difficulty in satisfying basic needs like food–was associated with recurrent stress. This has been shown to create cognitive loads that interfere with decision making in various ways beyond the financial constraints people may experience [16]. The following commentary clearly illustrates this:

> "... *You [can] see that there's no work and my husband has to figure out what we're going to eat tomorrow. And he worries because my son needs money for school and sometimes, he doesn't have work. And this is always a worry ...*" (Married woman, 26 years old, elementary education, housewife, No. 17).

**Table 4. Cognitive biases identified by level of intensity.** Oaxaca, México, 2019.

| Type of cognitive bias | Frequency |
|---|---|
| Cognitive load | +++ |
| Present bias and incentives | +++ |
| Social norms | ++ |
| Availability of information | ++ |
| Simplicity of processes and influence of those who provide information on the program | ++ |
| Loss aversion | ++ |
| Commitments | + |
| *Status quo* bias | + |

**Note:** Levels of frequency are expressed as +++ = High, ++ = Moderate, + = Low.

Financial stress and the perceived delay in enjoying the benefits of participating in the *MIDIT* were associated with an undervaluing of the importance of dedicating time to monitoring neurodevelopment or receiving information to improve child-rearing practices.

## Present bias and incentives

Participants recognized the potential benefits of greater involvement with the *MIDIT*. Nonetheless, they explained that the limited resources available for education and child-rearing led them to question the value of the basic food packages distributed by the program in the short, medium, and long terms. Some beneficiaries declared that their participation had declined since the suspension of food package deliveries. Such was the case of the following interviewee:

> "... I think so, right? Because as soon as they stopped giving [the nutritional package] they stopped coming around anymore and [participation] dropped a lot ..." (Group of married women, No. 5).

This greater appreciation for the food package reflected a heightened *present bias*, leading to a questioning of the value of the food package in comparison with the measuring and monitoring of infant physical and neurological development, the delivery of nutritional supplements, and the talks and workshops on health and child-rearing. These comments were most frequently heard from "demotivated" respondents who had abandoned the program. In contrast, those committed to the intervention showed greater ownership and appreciation of the help received in the areas of childhood health and development, as well as increased adoption of the concepts promoted by the *MIDIT* regarding encouraging healthy development. As the following interviewee put it:

> "... I brought my daughter because sometimes we don't have enough. What we earn simply isn't enough. And so, we came here because it helps us some ..." (Married woman, 28 years old, middle-school education, housewife, No. 15).

## Social norms

Among interviewees, we observed both descriptive (how people tend to behave) and prescriptive (acceptable behavior) social norms defining the roles of men and women. Many maintained a traditional view of domestic practices, roles, and child-rearing, with women in charge of childcare, home care and food preparation. Depending on the financial circumstances and

composition of their families, women also participated, to a lesser extent, in the pursuit of income. As described by the following interviewee:

*". . . [My main tasks as a woman] are preparing food, talking with the children to make sure they also eat. Sometimes the fathers are angry and attack them, instead of knowing how to [prepare things] on their own. We have to talk to them lovingly and not give them more anger or more problems. But with the children, no, we shouldn't take it out on them. It's not their fault . . ." (Married woman, 38 years old, elementary education, engaged in agricultural work and caring for the home, No. 19).*

Most men, on the other hand, identifying themselves as providers, did not report participating in child-rearing tasks beyond engaging in recreational activities with their children. They *did* however participate in decisions regarding spending on health and education. One father described his role in the family as follows:

*". . . the most important thing a father does for his children is to work so that they're not lacking, so that they have enough to eat . . ." (Married man, 29 years old, elementary education, farmworker, No. 12).*

Child-rearing skills proved an important factor in the decision to continue in or drop out of the program. Those committed to participating identified the following as important elements of educating and raising their children: play, being attentive to their health and demonstrating affection–elements well beyond satisfying basic needs like food. Respondents who had been involved longer with the program mentioned practicing the stimulation techniques acquired through the intervention, activities promoting childhood growth/development, continually demonstrating affection and, in general, providing a convivial environment in the home. In this group, mothers showed greater understanding and appreciation of *MIDIT* information and activities. In addition, these participants valued the monitoring of the physical health and neurodevelopmental status of their children. They also attached special importance to the improvement in child-rearing knowledge and skills gained through the *MIDIT*. In these cases, fathers reported playing a more active role in raising their children, a positive opinion of the intervention, and greater knowledge of program benefits than those who had left the program. The following commentary outlines the knowledge and skills acquired by the mothers:

*". . . a mother must have time for her children, to play with them, feed them good food, show them affection, right? Show them affection and make sure they're healthy . . ." (Widowed mother, 32 years old, high school education, dedicated to farming and housework, No. 1).*

## Availability of information

Involvement of fathers in the decision to participate in the program was related to their level of knowledge concerning the intervention and their inclination to reflect on aspects of child-rearing related to the development of their children. The level of knowledge on the part of fathers concerning the intervention was generally low. This manifested as poorly informed decisions regarding participating or continuing in the program, often based on incomplete information. In households disappointed in the program, fathers were less sensitive to the importance of monitoring the nutritional and neurological status of their children, as well as of improving their child-rearing skills. The following statement from fathers reflects their limited knowledge of program activities:

*". . . I know they come to town to deliver Un Kilo de Ayuda, and I know that someone comes to distribute it and plays games around here with. . .. with the mothers. Un Kilo de Ayuda, I think they hand out a bag of rice, beans [. . .] Well, here in the clinic [we took my boy] barely a week ago, I think. I don't think he, on Monday they just took him to Miahuatlan for a shot. I don't really remember what you're telling me about them weighing him. Yeah, they weighed him. They weighed him on Thursday for the same reason, that he was a little sick . . ." (Spouse of demotivated participant, male, 40 years old, finished middle school, farm and construction worker).*

Fathers who were more involved in the home and demonstrated greater knowledge of the intervention had fewer neutral or negative opinions of the intervention, as shown below:

*". . . I'm more or less informed: it's about the children's health, right? They're keeping an eye on them, their health, their health condition, that they're eating well. [It's about making sure the kids] aren't undernourished, yeah, on malnutrition. That's it, I don't have any more information. Since I haven't been able to go, this is what people've told me. So, when there are talks, she [my wife] comes home and we start talking about what they told her. In my case, that's fine. The most important thing in life is health, staying healthy. When there's health there's everything else. . ."*
*(Spouse of committed participant, man, 28 years old, finished middle school, farm/field worker).*

## Simplicity of processes and influence of program staff who provide the information

Other factors that influenced the decision to participate, continue with or abandon the program included the following: organizational characteristics and operational processes, in particular the ease of registration (which mitigated the limited attention biases present in contexts marked by deprivation), the sufficiency and relevance of the benefits offered, and the way participants were treated by staff. No barriers were reported by *UKA* program participants regarding enrollment requirements or the registration process. The following commentary confirms this:

*". . . It was easy [to enroll my son in the program]. After we enrolled him, in eight or fifteen days we had the first talk. The package came later, after we'd had about two talks . . ." (Focus group interview with a committed participant, No.1).*

The program outreach strategy consisted primarily of awareness-raising talks with community members before their deciding whether or not to participate. Some accounts demonstrated that understanding and adoption of *MIDIT* concepts increased with the amount of time mothers spent in the program. The following account and other commentaries from demotivated participants revealed a positive perception concerning the reputation of the program and the organization:

*". . . because they said that it was a good program where, besides distributing food, they would teach us how to feed our children, what types of food it was better to give them if they detected they had anemia . . ." (Participant from the group of married women, No. 27).*

## Loss aversion

The value families accorded to financial benefits in relation to the costs they would incur by participating in the program stood out. For example, in rural communities where homes were

far from the meeting venues for the distribution of goods and services, families systematically calculated all the costs of attending program events, such as transportation costs and the activities they would have to forgo. The following opinion represents the favorable attitude of interviewees towards the financial benefits of the program:

> "... They give things at a low price. If we had to buy the food they give us, we'd spend more money. So, with what they give us, money lasts more. So I say that's why people come around, right? Because they can buy these products at a lower price ..." (Married woman, housewife, Focus group interview, No. 37).

While deciding whether to continue with or abandon the program, some families gave serious consideration to the consequences of losing the program incentives; they were aware of the gains they had acquired and compared their current status with the situation that would ensue if they left the program. Worthy of note were the commentaries of caregivers who were aware of the non-discount monetary value not only of the food package incentive in stores, but also of the medical tests and nutritional treatments provided for their children. We observed that caregivers who most valued the diagnostic assistance, care and follow up given their children took their current monetary benefits and the risk of losing them into account more frequently. As expressed by the following interview:

> "... My mother-in-law has asked me several times why I come here. "They don't give you anything," she says, but I reply: "Apparently they don't give anything, but they're taking care of my child and I don't want to lose that..." (Woman, single mother, 35 years old, elementary education, farm worker and housewife, No. 01).

## Commitments

The presence of problems related to childhood malnutrition and its prevention was also an important factor in the decision of parents to participate and remain in the program. During general meetings with community members, we observed an emerging interest from potential beneficiaries in participating with the purpose of obtaining diagnoses of anemia and/or underweight for their children. The following statement exemplifies this:

> "... Also, something really important they commented to me was that they would monitor the children better than at the health center. That's why I decided to enroll my son ..." (Participant from the group of married women, No. 27).

## *Status quo* bias

Families with access to, or knowledge of, other quality social programs and health services aimed at improving and monitoring the nutrition and overall health of their children valued MIDIT benefits to a lesser degree. Staying with the program notwithstanding the possibility of benefiting from other interventions they were familiar with and perceived as effective denoted their preference for what was currently in place. However, their decision to remain may have also reflected the inability to attend two care options competing for their limited available time. Other participants also referred to a time conflict between participating in social programs and handing their domestic activities, leading to a greater commitment to remaining in the program to improve the nutritional status of their children. This point is illustrated below:

*" . . . Well, it also depends on where the community's located because you have a community—it's what I see most—you have communities that are very near an urban zone, and they're the ones that go least, because they don't go. It's the communities that go–eight, nine, twelve, fifteen–the communities that are obviously farther away; they're the ones that go: fifty, sixty. That's why I, it's like they say, you struggle to get to that community, no matter what the road looks like, you know you're going to get there because all the ladies will be there, whether or not there's a package, they're going to go. Then it's also going to depend on the services that the community has. It depends a lot on that, if the community really has services. In other words, really, for you, like, not really, because I can do that at the health center, or the nurse does that. It's like, what am I going for . . .?" (Focus group interview with UKA Facilitators, No. 1).*

## Cognitive biases identified by level of intensity

By way of summary, Table 4 presents the degree of intensity of the biases identified. Our analysis suggests that *cognitive load* and *present bias* were the biases that most severely affected the decision-making capacity of beneficiaries. The first generated cognitive stress, limiting attendance and participation, while the second led to an overvaluing of immediate as opposed to medium- and long-term program benefits. The latter created a dependence on the food packages as an incentive among both recipients and facilitators. We assigned them the highest value as they proved the principal determinants of the decision to either continue with or abandon the program.

We assigned an intermediate value to social norms, availability of information, simplicity of program processes, *loss aversion* and influence of those administering the programs. These were regarded as having less influence on the decisions of beneficiaries, in spite of influence from the fathers which, tended to discourage participation, along with the underlying pressures of the social context.

The lowest values pertained to the *status quo* bias and commitment, owing to the difficulties we observed among women in translating the effects of ECD intervention to their children. Notwithstanding their occurrence among participants, these factors were not cited as dissuasive in the decision to suspend attendance.

## Determinants of participation and/or abandonment

Fig 3 presents the factors that shaped the decision to participate in or abandon the analyzed program and their interaction. The attitude of the mothers towards the intervention was closely related to their knowledge of the intervention, a consequence of their growing familiarity with ECD program benefits. It was also related to their involvement in the monitoring of the health of the children, and to their favorable opinion of the intervention. Field observations suggest that the resilience of the mothers played a key role, above all for their engagement in monitoring the growth indicators of their children. On the domestic front, the involvement of the fathers in childcare and their having a favorable opinion of the intervention were positive determinants of participation in the program.

Factors influencing the decision to abandon the program are presented in Fig 4. We can see that the principal determinants among the least committed mothers were the time and effort they were required to invest in program activities, as well as a lower level of appreciation of the intervention. These factors may have played a larger role in discouraging participation for those mothers less engaged in childcare, or when their partners dampened their enthusiasm, questioning the immediate benefits of their involvement.

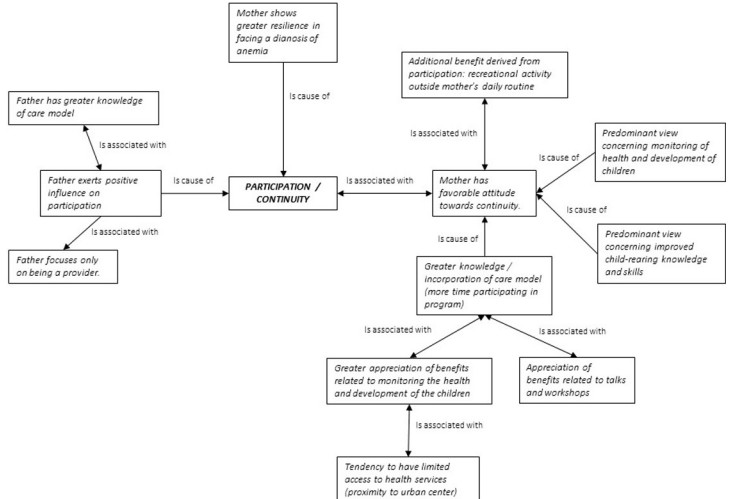

**Fig 3. Factors related to participation / continuity in the ECD programs, Oaxaca, 2019. Source**: Elaborated by the author using Atlas-Ti, V7 software [56].

## Typology of participation

The profiles of the women who decided to participate and remain in the ECD program are presented in Fig 5. The predominant characteristic observed was time of exposure to the care model, followed by sensitive childcare, concern with childhood anemia and the participation of fathers in childcare. These attributes were complemented by the relevance attached by the women to the role of the *UKA* organization. Their appreciation was influenced by a favorable opinion of the intervention on the part of the fathers, and by the possibility of gaining additional knowledge of the benefits offered by the program.

Meanwhile, the principal factors limiting participation included less exposure to the care model, high levels of household financial stress, and an overvaluing of financial incentives. To a lesser extent, participation was also deterred by the non-involvement of fathers in childcare,

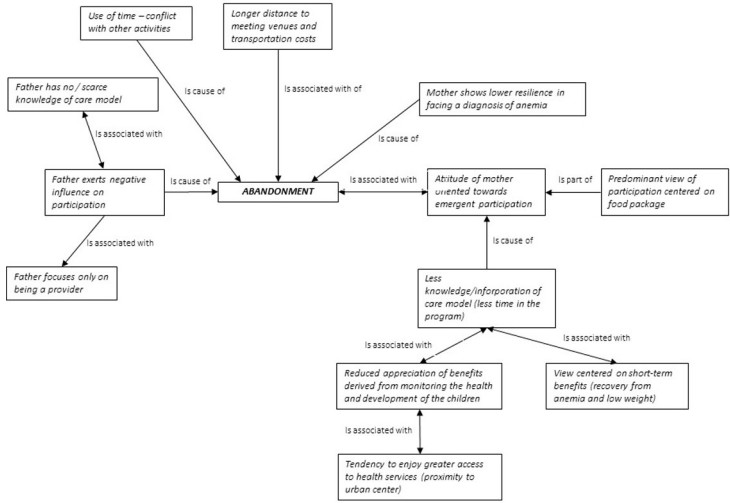

**Fig 4. Factors related to abandonment of ECD programs, Oaxaca, 2019. Source:** Elaborated by the author using Atlas-Ti, V7 software [56].

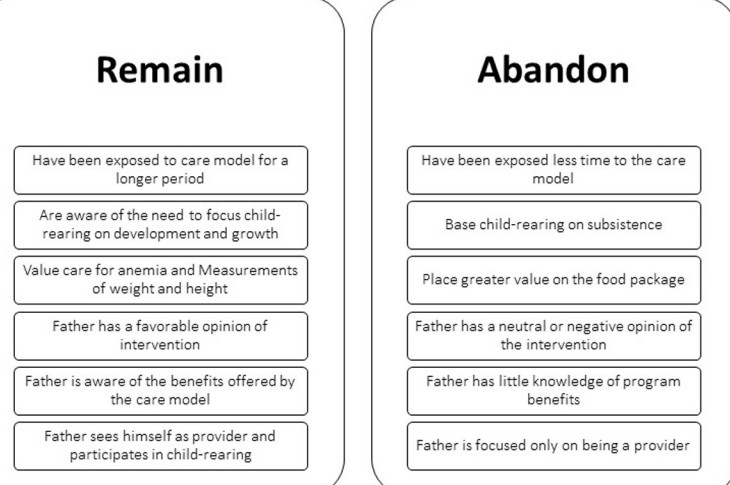

**Fig 5. Individual profiles concerning participation in ECD program. Source:** Elaborated by the authors according to the findings of the study.

their low assessment of the intervention and the unawareness of program benefits on the part of mothers.

## Discussion and conclusion

Based on conceptual elements from the field of Behavioral Economics (BE), this study contributes original evidence on the cognitive biases that influence the decision of potential beneficiaries to participate and remain in ECD interventions in poor rural areas in Mexico. Beyond the bounds of conventional theory, our analysis considered the possibility that people are not concerned about their long-term well-being because they are more worried about the present. Our results on the interaction between contextual determinants and cognitive biases converge with the BE stance regarding the need to understand these biases in order to increase the levels of participation in social programs and bring about greater impact for ECD interventions [11, 12, 14, 21].

In line with the literature reviewed as part of our study, our findings show that the experience of deprivation interferes with rational decision making within households in that it obscures the perception of program benefits and thereby affects the decision to remain in or abandon social interventions [22]. Our field data revealed that insecurity concerning basic necessities such as food manifested as recurrent stress, creating additional cognitive loads that hindered decision making. In this context, we confirmed that financial stress played a key role in the decisions to participate and continue with the program of interest. It generated a response denominated *tunnel vision effect* in which the search for solutions to the most pressing current problems impeded appreciating the effects of the intervention over the medium- and long-terms.

The benefits for families of interventions such as those reviewed are often not felt immediately. This was the case with several of the ECD activities analyzed such as monitoring neuro-development and providing information to improve child-rearing practices. The evidence gathered confirms that financial incentives function as "nudges" with a domino effect. They promote initial participation rooted in the perception of immediate benefits and, in turn, lead to changes favoring medium-term improvement in the health, nutrition and development of children, the ultimate beneficiaries of ECD programs. As shown in the literature, short-term

financial incentives serve as a catalyst for the pursuit of long-term benefits such as appropriate neurodevelopment [58].

In the case of Mexico, one contextual element that has been widely studied and deserves special attention is the extensive delivery of food packages to various populations by public and private programs between 1994 and 2018. The resulting oversupply of nutritional services indicated the widespread acceptance of many public programs until their suspension by federal and state governments. In light of this, the reputation of NGOs, benefit "priming" and the ways in which information is framed for beneficiaries have become key elements in maintaining participation in programs such as the ECD intervention analyzed. These elements help assure that any decline in the assessment of these programs is not because of a lack of clarity regarding their benefits [19]. Another useful proposal related to program design is the need to accord greater consideration to the time community members are required to invest to take advantage of the available care options. Finally, various authors have outlined the conditions under which incentives can make the difference [15, 21, 39]. Among them, we observed that the influence of those implementing the intervention was key in the decisions of participants to remain. Implementers must be sensitive to the fact that living in impoverished circumstances constrains the attention parents are able give to the project. The use of simple processes can help diminish the level of fatigue in decision making. Thus, the simplicity and relevance of the information provided to caregivers can guarantee that messages are being adequately understood.

Our results indicated that some families undertook cost/benefit assessments of the expenses they incurred by continuing to participate in the intervention, while expressing an aversion to loss [30]. This attitude was particularly prevalent where fathers exercised greater influence over domestic decision making, a factor favoring the abandonment of the program. Our observations are consistent with those of previous studies as regards the need for social programs to provide information and incentives that engage men–normally ignored when designing child interventions–and enhance the role of women in domestic decision making [33]. Our data also suggests that the decision of whether or not to participate in a program is often made on the basis of incomplete information, allowing cognitive biases to expand their scope. In households that abandoned the program, fathers demonstrated less awareness of *MIDIT* benefits. It has been demonstrated, however, that as interventions advance, mechanisms are created among the population that tend to boost the positive assessment of the programs on the part of beneficiary families.

Our research had several limitations. First, our findings cannot be generalized to other contexts where similar interventions are under way. Second, several observed sociocultural elements that affected domestic decision making are currently under significant social pressure, generating changes in the medium term that may remove several barriers. Finally, the qualitative approach and data collection strategy used may have led to bias in the observations of the field staff. Additional data source triangulation would have lent greater rigor to our analysis.

Notwithstanding these limitations, however, we believe that BE endeavors such as ours can contribute to improving the design of social programs in contexts of social and political transition like those occurring in Mexico and other Latin American countries. In the face of financial vulnerability, it is crucial to become familiar with cognitive biases in local communities in order to anticipate events that could put program effectiveness at risk. In addition, it is essential to apply BE research results in order to create conditions that sharpen service provision procedures, thus allowing ECD programs to maximize benefits to participants. Greater attention to *risk aversion* and better management of incentives can guide the decisions of policy makers when deciding to undertake interventions that require individuals to make decisions in their best interest.

## Acknowledgments

We thank the study participants for their valuable contributions. We are also especially grateful to Blanca Laura Ortega Román for her role as the general coordinator and research assistant of the project.

**Memorial dedication**

We dedicate this paper to our colleague Dr. Sandra Sosa-Rubi who passed away in March 2021. Dr. Sosa will be remembered as a remarkable health economist, friend, and human being.

## Author Contributions

**Conceptualization:** Edson Serván-Mori, Carlos Pineda-Antúnez.

**Formal analysis:** Edson Serván-Mori, María L. Bravo-Ruiz, Mariana Molina, Martín I. Ramírez-Baca.

**Funding acquisition:** Edson Serván-Mori.

**Investigation:** Edson Serván-Mori, Mariana Molina, Martín I. Ramírez-Baca, Angélica García-Martínez, Amado D. Quezada-Sánchez, Emanuel Orozco-Núñez.

**Methodology:** Edson Serván-Mori, María L. Bravo-Ruiz, Emanuel Orozco-Núñez.

**Software:** Emanuel Orozco-Núñez.

**Supervision:** Edson Serván-Mori.

**Writing – original draft:** Edson Serván-Mori, Carlos Pineda-Antúnez, María L. Bravo-Ruiz, Mariana Molina, Martín I. Ramírez-Baca, Angélica García-Martínez, Amado D. Quezada-Sánchez, Emanuel Orozco-Núñez.

**Writing – review & editing:** Edson Serván-Mori, Carlos Pineda-Antúnez, Mariana Molina, Martín I. Ramírez-Baca, Angélica García-Martínez, Amado D. Quezada-Sánchez, Emanuel Orozco-Núñez.

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
