## [Decision Letter · Decision Letter 0]

4 Mar 2021

PONE-D-20-39705

A behavioral economics analysis of the participation in early childhood development social programs promoted by civil societies in Mexico

PLOS ONE

Dear Dr. Emanuel Orozco-Núñez,

Thank you for submitting your manuscript to PLOS ONE. After careful consideration, we feel that it has merit but does not fully meet PLOS ONE’s publication criteria as it currently stands. Therefore, we invite you to submit a revised version of the manuscript that addresses the points raised during the review process.

I have had a chance to read your paper and heard back from 2 referees. The enclosed referee reports provide useful feedback. While the second referee likes the point of the paper and the fact that it was well-written, the first referee points out some concerns.

Based on the referees' recommendations and my own review, I kindly ask you to submit an improved version. Please follow the referees' recommendations and questions and make sure you respond to them in detail.

We look forward to receiving your revised manuscript.

Kind regards,

Mosi Rosenboim

Academic Editor

PLOS ONE

Journal Requirements:

"This work was possible with the support of the Non-governmental Organization Un

Kilo de Ayuda A.C., Mexico (UKA). The funder was not involved in the study design or data

collection and had no say in the decisions related to data analysis or interpretation"

"No. The funders had no role in study design, data collection and analysis, decision to publish, or preparation of the manuscript."

Reviewers' comments:

Reviewer's Responses to Questions

**Comments to the Author**

1. Is the manuscript technically sound, and do the data support the conclusions?

Reviewer #1: Partly

Reviewer #2: Yes

2. Has the statistical analysis been performed appropriately and rigorously? 

Reviewer #1: N/A

Reviewer #2: Yes

3. Have the authors made all data underlying the findings in their manuscript fully available?

Reviewer #1: Yes

Reviewer #2: No

4. Is the manuscript presented in an intelligible fashion and written in standard English?

Reviewer #1: Yes

Reviewer #2: Yes

5. Review Comments to the Author

Reviewer #1: PONE-D-20-39705

A behavioral economics analysis of the participation in early childhood development social programs promoted by civil societies in Mexico

Comments to Author

Thank you for the opportunity to review the work entitled "A behavioral economics analysis of the participation in early childhood development social programs promoted by civil societies in Mexico". The current study uses the behavioral economics approach, to analyze the decision to participate in an early childhood development program implemented in Mexico by a nongovernmental organization.

This paper focuses on an interesting topic; however, the paper has some areas for improvement and clarification. Please see my comments below.

Introduction

I think you should highlight more clearly, how this research will contribute to both theory and practice. Overall, the Introduction fails to present a cogent rationale for the study.

It is recommended to restructure the text so that it provides a better lead-in into the current inquiry.

Literature review

- Several empirical studies have explained the ways in which BE has influenced the design of Policies, see for example:

Axelrad, H., Luski, I., Malul, M. (2016). Behavioral Biases in the Labor Market Differences between Older and Younger Individuals. Journal of Behavioral and Experimental Economics. 60(1), Feb. 2016, 23-28. doi: http://dx.doi.org/10.1016/j.socec.2015.11.003

- The theoretical contribution is not obvious. How the current research will provide new insights? Where does the paper add to the vast body of knowledge?

- In addition, the literature review should give some research questions. Now it is not clear what questions are addressed with the proposed method. In this context the conclusion should be changed and should refer to those research questions.

Method

The section on methodology created more confusion than clarity for me:

- The criteria of the selection of the interviewed people are not explained.

- When talking about sampling, can you explain your sampling method? Was it snow balling? Purposive sampling?

- The description of the sample is not clear. Please provide more information; you might want to add a table with a description of the interviewees and focus group.

- The section on methodology further needs to clarify what the diversity of the sample means in terms of scope and limitations.

- It is also a good practice to justify why the specific methodology is used.

Results

- The findings section is a bit imprecise in detailing how many respondents brought forward similar arguments.

For example: “Various participants recognized….”, “Among interviewees, we observed several both descriptive….” “Most men, on the other…”

- The claim that behavioral economics has an impact on the decision to participate in an early childhood development program is supported by the minutes of the interviews, with no other test or empirical analysis. This leaves the reader with the impression that the research is mostly anecdotal.

Discussion and Conclusions

- In the discussion section, the findings should again be linked with previous literature with similar results this will help substantiate your theoretical contributions.

- The sections on limitations, implications and future research are underdeveloped.

- Currently the contribution of this research to the field of behavioral economics is not clear.

I hope you find my comments helpful. Good luck with the further development of your paper.

Reviewer #2: This research discusses how the conceptual components of behavioral economics can be used to analyze the decisions of residents of poor communities in Oaxaca (Mexico) to participate and remain in or abandon ECD programs implemented by the Mexican non-governmental organization. The authors identified eight biases associated with these decisions: 1) cognitive load; 2) present bias and incentives; 3) social norms; 4) availability of information; 5) simplicity of the process and 10 influence of intervention facilitators; 6) loss aversion; 7) commitments and 8) status quo bias. The authors concluded that this kind of research can contribute to improving the design of social programs.

Evaluation

Both the question posed and the findings reported in the paper are interesting. The paper is well-polished and has a worthy contribution of determining the conceptual components of behavioral economics that can be used to analyze the decisions of residents to participate in social programs. Overall, the paper is well-suited to be published in Plos One.

Comments

1. The authors mentioned in page 3: "The importance of incorporating the BE perspective into the design and implementation of social programs has been widely recognized." Several examples have been provided, but the authors should expand on the contribution of behavioral economics to these programs.

2. The participants were divided to three groups: "commissioners", "demotivated" and the third group included managers, coordinators and facilitators. In the analysis of the results there was almost no reference to the differences between the groups. Has this analysis been performed? Were there any differences? It is worth noting the differences, if any.

3. The main emphasis in behavioral economics is to investigate the cases in which the behavior of individuals is not always compatible with the usual economic assumptions of perfect rationality and decision making according to personal interests. The terms of rationality and irrationality are not mentioned at all in the article. I suggest addressing irrationality (by economic definition) in describing the concepts relevant to the decision making of program participants.

4. The connection between loss aversion and the results is discussed (page 14), but not in a clear and satisfactory manner. It is worth rephrasing and expanding.

5. Continuing from the previous comment, the same goes for the status quo bias (page 15).

6. It is not clear how Table 3, Figure 3 and Figure 4 were prepared, please detail clearly. What is the relevant data? For example, how the authors determine the level of intensity of each cognitive bias?

7. The authors can cite additional relevant papers, such as:

Bounded Rationality:

Greenberg, D. ,et al. (2016). Can Financial Education Extend the Border of Bounded Rationality? Modern Economy, 7, 103-108.

Present Bias:

Bayer, Y., et al. (2019). Time and risk preferences, and consumption decisions of patients with clinical depression. Journal of behavioral and experimental economics, 78, 138-145.

Present Bias:

Shtudiner, Z. (2018). Risk Tolerance, Time Preference and Financial Decision-Making: Differences between Self-Employed People and Employees. Modern Economy, 9, 2150-2163.

6. PLOS authors have the option to publish the peer review history of their article (what does this mean?). If published, this will include your full peer review and any attached files.

Reviewer #1: No

Reviewer #2: No

---

## [Author Response · Author response to Decision Letter 0]

16 Nov 2021

Comments from the Editor:

RESPONSE: Thank you for your recommendation. Text and figures were adjusted as indicated. In order to have a more specific article title, we also decided to modify it as follows: Participation in early childhood development programs promoted by civil societies in Mexico: a behavioral economics analysis.

RESPONSE: Thank you for this suggestion. We consider important to provide additional information; as we conducted qualitative field research, we agreed and signed informed consent in all the cases. As part of this procedure, we explained to our informants that audio tapes were erased after transcription and that only the research team could read the interviews. Fieldwork and interviews were conducted in Spanish language using four specific guides and a manual. In order to respond to your request, we offer access to qualitative data base and field instruments in Spanish versions. It is going to be a hard labour to translate these materials to be presented as documentary support.

Reviewer #1:

1. Introduction. I think you should highlight more clearly, how this research will contribute to both theory and practice. Overall, the Introduction fails to present a cogent rationale for the study. It is recommended to restructure the text so that it provides a better lead-in into the current inquiry.

RESPONSE. Thank you. We have made the requested changes.

2. Literature review. Several empirical studies have explained the ways in which BE has influenced the design of Policies, see for example: [Axelrad, H., Luski, I., Malul, M. (2016). Behavioral Biases in the Labor Market Differences between Older and Younger Individuals. Journal of Behavioral and Experimental Economics. 60(1), Feb. 2016, 23-28. doi: http://dx.doi.org/10.1016/j.socec.2015.11.003]. The theoretical contribution is not obvious. How will the current research provide new insights? Where does the paper add to the vast body of knowledge? In addition, the literature review should give some research questions. Now it is not clear what questions are addressed with the proposed method. In this context the conclusion should be changed and should refer to those research questions.

RESPONSE. We thank the reviewer for this comment and would like to clarify our propose as follows: […] Despite research and interest in behavioral economics and psychology have grown by leaps and bounds in the last few decades, an important gap in the available literature with respect to the decision of community members to participate and remain in the social programs for early childhood development (ECD) programs 1,2. Moreover, there is little evidence regarding the understand the behavioral correlates to them in high social deprived contexts in LMICs as Mexico. This is particularly relevant if we consider that being a baby, girl or boy is an additional factor of social vulnerability to those who face these contexts, because the enforceability of their rights depends on the decision and preferences of other people. Based on a brief literature review on the connection between BE and the design of social programs, and after to review the conceptual components of BE can be used to analyze the decisions of individuals to participate and remain in or abandon social programs, this paper aims to analyze the decisions of residents of poor communities in Oaxaca, in Mexico, who participated in the ECD programs implemented by the Mexican non-governmental organization Un Kilo de Ayuda (UKA). Finally, this work ponders the implications of our findings for the design, implementation, monitoring, and performance of ECD programs […]

3. Method. The section on methodology created more confusion than clarity for me: -The criteria of the selection of the interviewed people are not explained. - When talking about sampling, can you explain your sampling method? Was it snow balling? Purposive sampling? - The description of the sample is not clear. Please provide more information; you might want to add a table with a description of the interviewees and focus group. -The section on methodology further needs to clarify what the diversity of the sample means in terms of scope and limitations. -It is also a good practice to justify why the specific methodology is used.

RESPONSE. We reviewed and adjusted the section of study site and participants in the Material and methods section. From the first lines of page nine we clarified that we conducted a purposive sampling and clarified the number and type of interviews by presenting a table that describes the type and number of interviews and participants.

4. Results. - The findings section is a bit imprecise in detailing how many respondents brought forward similar arguments. For example: “Various participants recognized….”, “Among interviewees, we observed several both descriptive….” “Most men, on the other…” - The claim that behavioral economics has an impact on the decision to participate in an early childhood development program is supported by the minutes of the interviews, with no other test or empirical analysis. This leaves the reader with the impression that the research is mostly anecdotal.

RESPONSE. We appreciate this observation, which is recurrent while reading qualitative data. In order to provide a satisfactory answer to this requirement, we reviewed and corrected vague arguments, trying to provide a more direct argument about the described cognitive biases. Additionally, we expect that the description of our conceptual framework, interview guides and validity and rigour provide elements of our effort to avoid anecdotal situations. We were careful providing additional data on our presentations, such as field observations.

5. Discussion and Conclusions. - In the discussion section, the findings should again be linked with previous literature with similar results this will help substantiate your theoretical contributions. - The sections on limitations, implications and future research are underdeveloped. - Currently the contribution of this research to the field of behavioral economics is not clear.

RESPOSE. We thank this observation, because it helped us to focus on the behavioural economics approach and its implications in our results. On the last paragraph of p. 17, we eliminated some confusing lines and focus on comparing and analysing implications of our research with the literature on BE. Finally, on the last paragraph we stated in more clear way the role of BE approach for improving ECD programmes.

Reviewer #2:

1. The authors mentioned in page 3: "The importance of incorporating the BE perspective into the design and implementation of social programs has been widely recognized." Several examples have been provided, but the authors should expand on the contribution of behavioral economics to these programs.

RESPONSE. We thank the reviewer for this comment and would like to clarify the following: […] The importance of incorporating the BE perspective into the design and implementation of social programs has been widely recognized. For instance, the project Behavioral Interventions to Advance Self-Sufficiency (BIAS) in the United States has demonstrated the benefits of implementing child-centered policies by incorporating “nudges” regarding the well-being of children. The authors of this project described the cost-effectiveness of considering biases such as social influence, personalization and loss aversion when designing a child-care intervention. Their results indicated improvements of two-to-four percentage points in the indicators analyzed, namely keeping medical appointments, establishing commitments with social assistance programs and requesting educational credits during appointments 3. Other studies have shown that using BE elements such as present bias, temporal preferences, loss aversion and context can accelerate improvements in maternal and child health4. The review of programs or interventions, from a BE point of view, made it possible to identify the precise moments in which it is necessary to provide assistance to parents for better decision-making, identify improvements in the time, content and form communication with parents, establishing reminders of important phases, as well as simplifying procedures. Specifically, behavioral economics allows us to identify the improvements that can support better decision-making, by recognizing that people make decisions with limited information and not necessarily in the best moments and contexts. […]

2. The participants were divided to three groups: "commissioners", "demotivated" and the third group included managers, coordinators and facilitators. In the analysis of the results there was almost no reference to the differences between the groups. Has this analysis been performed? Were there any differences? It is worth noting the differences, if any.

RESPONSE. The analysis was made on the base of the differences between the point of view of the “committed” parents and the “demotivated” parents. The information obtained from the third group gave context on the participation of the first two groups and on the operational aspects that could affect this participation. The description of the groups of participants was re-phrased in order to clarify the differences. We revised the section to improve its clarity, as follows: […] For our study, we selected four localities with high and very high levels of social marginalization served by the UKA organization: San Antonino el Alto, Magdalena Yodocono de Porfirio Diaz, Rancho San Felipe (Santiago Matatlán) and San Simon Almolongas. Located 30-70 kilometers from the state capital, the inhabitants of these towns are engaged primarily in the production and distribution of agricultural products, livestock raising, retail trade, construction work and domestic work. From May 20 to 30, 2019, we conducted 30 semi-structured interviews and seven focus groups with a total of 61 participants (51 women and 10 men) divided into three large groups. The first group consisted of the “committed” parents of the children benefited by the program, who believed in the importance of participating in the MIDIT, this group included the “commissioners,” committed mothers who also worked as translators/interpreters and participated in UKA operations and their partners (n=12). The second group corresponded to the “demotivated” parents, who had participated at some point and then abandoned the program (n=7 in total). The third group was not a group of beneficiaries, but it included the program operating staff: the manager of the CEDIT, the coordinator of the PDNyP and the facilitators of MIDIT activities (n=8 in total). This group was included in the analysis in order to give context to the participation of the beneficiaries and to obtain information on the operational processes that could affect said participation. […]

3. The main emphasis in behavioral economics is to investigate the cases in which the behavior of individuals is not always compatible with the usual economic assumptions of perfect rationality and decision making according to personal interests. The terms of rationality and irrationality are not mentioned at all in the article. I suggest addressing irrationality (by economic definition) in describing the concepts relevant to the decision making of program participants.

RESPONSE. Following the suggestion, we have expanded the paragraph about irrational decisions by adding more detail as follows: […] Neoclassical microeconomic theory assumes that individuals are fully informed, make optimal and rational decisions, have well-defined preferences and calculate choices rapidly before making a decision 5. However, these assumptions are not always correct. Since the advent of BE, a growing body of evidence has indicated that cognitive biases are often associated with suboptimal decision making. Irrational decisions in the economy are decisions that do not maximize utility and provokes a loss of economic welfare. One of the main emphasis in behavioral economics is to explore why people make irrational decisions and why their behavior does not follow the economic models explaining the association between the dominant choice for most people in society and the cognitive bias 6 […]

4. The connection between loss aversion and the results is discussed (page 14), but not in a clear and satisfactory manner. It is worth rephrasing and expanding. Continuing from the previous comment, the same goes for the status quo bias (page 15).

RESPONSE. Both sections were re-phrased.

6. It is not clear how Table 3, Figure 3 and Figure 4 were prepared, please detail clearly. What is the relevant data? For example, how the authors determine the level of intensity of each cognitive bias?

RESPONSE. Levels of intensity were defined considering the frequency and detail of the described cognitive biases. Usage of Atlas-Ti was very helpful to have a precise idea because the software registers the number of occasions in which a code was used. At the same time, this software has functions to define the type of relations that could be defined among the different used codes. For example, in Table 4 we established that cognitive load had much more analytical references, comparing with status quo bias, which was not so frequently expressed and detailed. Relationships among analysed topics defined in diagrams considered the way in which cognitive biases were expressed. In this case, the way in which the discourse of mothers expressed cognitive biases, usually combined some of these biases; we considered this combination to define the type of relationships presented in the diagrams.

7. The authors can cite additional relevant papers, such as: Bounded Rationality: [Greenberg, D., et al. (2016). Can Financial Education Extend the Border of Bounded Rationality? Modern Economy, 7, 103-108.] Present Bias: [Bayer, Y., et al. (2019). Time and risk preferences, and consumption decisions of patients with clinical depression. Journal of behavioral and experimental economics, 78, 138-145.] Present Bias: [Shtudiner, Z. (2018). Risk Tolerance, Time Preference and Financial Decision-Making: Differences between Self-Employed People and Employees. Modern Economy, 9, 2150-2163.]

RESPONSE. Thank you for the suggestion of these relevant papers. We included the suggested refences for present bias. […] Other studies have shown that using BE elements such as present bias, temporal preferences, loss aversion and context can accelerate improvements in maternal and child health 4,7,8. The review of programs or interventions, from a BE approach, made it possible to identify the precise moments in which it is necessary to aid parents for better decision-making, identify improvements in the time, content, and form communication with parents, establishing reminders of important phases, as well as simplifying procedures. Specifically, BE allows us to identify the improvements that can support better decision-making, by recognizing that people make decisions with limited information and not necessarily in the best moments and contexts. […]

---

## [Decision Letter · Decision Letter 1]

2 Mar 2022

A behavioral economics analysis of the participation in early childhood development social programs promoted by civil societies in Mexico

PONE-D-20-39705R1

Dear Dr. Orozco-Núñez,

We’re pleased to inform you that your manuscript has been judged scientifically suitable for publication and will be formally accepted for publication once it meets all outstanding technical requirements.

Kind regards,

Mosi Rosenboim

Academic Editor

PLOS ONE

Additional Editor Comments (optional):

Reviewers' comments:

Reviewer's Responses to Questions

**Comments to the Author**

1. If the authors have adequately addressed your comments raised in a previous round of review and you feel that this manuscript is now acceptable for publication, you may indicate that here to bypass the “Comments to the Author” section, enter your conflict of interest statement in the “Confidential to Editor” section, and submit your "Accept" recommendation.

Reviewer #1: (No Response)

Reviewer #2: All comments have been addressed

2. Is the manuscript technically sound, and do the data support the conclusions?

Reviewer #1: Yes

Reviewer #2: Yes

3. Has the statistical analysis been performed appropriately and rigorously? 

Reviewer #1: Yes

Reviewer #2: Yes

4. Have the authors made all data underlying the findings in their manuscript fully available?

Reviewer #1: No

Reviewer #2: No

5. Is the manuscript presented in an intelligible fashion and written in standard English?

Reviewer #1: Yes

Reviewer #2: Yes

6. Review Comments to the Author

Reviewer #1: Thank you for the opportunity to review the work entitled "A behavioral economics analysis of the participation in early childhood development social programs promoted by civil societies in Mexico".

Page 16: the sentence “Other participants also referred to a time conflict between participating in social programs and handing their domestic activities,

leading to a greater commitment to remaining in the program to improve the nutritional status of their children. This point is illustrated below:” appears twice

I have no further comments.

Reviewer #2: I'm happy with the effort of the authors in the revision.

They have addressed all my concerns.

7. PLOS authors have the option to publish the peer review history of their article (what does this mean?). If published, this will include your full peer review and any attached files.

Reviewer #1: No

Reviewer #2: No

---

## [Editor Report · Acceptance letter]

18 Mar 2022

PONE-D-20-39705R1 

A behavioral economics analysis of the participation in early childhood development social programs promoted by civil societies in Mexico 

Dear Dr. Orozco-Núñez:

I'm pleased to inform you that your manuscript has been deemed suitable for publication in PLOS ONE. Congratulations! Your manuscript is now with our production department. 

Kind regards, 

on behalf of

Dr. Mosi Rosenboim 

Academic Editor

PLOS ONE